# Radiomics and Clinicopathological Characteristics for Predicting Lymph Node Metastasis in Testicular Cancer

**DOI:** 10.3390/cancers15235630

**Published:** 2023-11-29

**Authors:** Catharina Silvia Lisson, Sabitha Manoj, Daniel Wolf, Christoph Gerhard Lisson, Stefan A. Schmidt, Meinrad Beer, Wolfgang Thaiss, Christian Bolenz, Friedemann Zengerling, Michael Goetz

**Affiliations:** 1Department of Diagnostic and Interventional Radiology, University Hospital of Ulm, Albert-Einstein-Allee 23, 89081 Ulm, Germany; sabitha.manoji@uni-ulm.de (S.M.); daniel.wolf@uni-ulm.de (D.W.); christoph.lisson@uniklinik-ulm.de (C.G.L.); stefan.schmidt@uniklinik-ulm.de (S.A.S.); meinrad.beer@uniklinik-ulm.de (M.B.); wolfgang.thaiss@uniklinik-ulm.de (W.T.); michael.goetz@uni-ulm.de (M.G.); 2ZPM—Center for Personalized Medicine, University Hospital of Ulm, Albert-Einstein-Allee 23, 89081 Ulm, Germany; 3XAIRAD—Artificial Intelligence in Experimental Radiology, University Hospital of Ulm, Albert-Einstein-Allee 23, 89081 Ulm, Germany; 4Visual Computing Group, Institute of Media Informatics, Ulm University, James-Franck-Ring, 89081 Ulm, Germany; 5MoMan—Center for Translational Imaging, Department of Internal Medicine II, University Hospital of Ulm, Albert-Einstein-Allee 23, 89081 Ulm, Germany; 6i2SouI—Innovative Imaging in Surgical Oncology Ulm, University Hospital of Ulm, Albert-Einstein-Allee 23, 89081 Ulm, Germany; christian.bolenz@uniklinik-ulm.de (C.B.); friedemann.zengerling@uniklinik-ulm.de (F.Z.); 7Department of Nuclear Medicine, University Hospital of Ulm, Albert-Einstein-Allee 23, 89081 Ulm, Germany; 8Department of Urology, University Hospital of Ulm, Albert-Einstein-Allee 23, 89081 Ulm, Germany; 9DKFZ—German Cancer Research Center, Division Medical Image Computing, 69120 Heidelberg, Germany

**Keywords:** radiomics, prediction, lymph node metastasis, testicular cancer, artificial intelligence

## Abstract

**Simple Summary:**

Testicular germ cell tumours (TGCTs) are the most common type of solid cancer in men under the age of 40. Of metastases from TGCTs, 95% involve the ipsilateral retroperitoneal lymph nodes. For early-stage TGCTs, the optimal treatment remains controversial, with options including surveillance, chemotherapy or lymph node surgery after orchiectomy. However, the accurate prediction of retroperitoneal lymph node metastasis is crucial to avoid unnecessary treatment and health complications in this group of young patients, highlighting the importance of precise follow-up care. In this study, we developed and validated predictive machine learning models integrating radiomics and clinical features for individual preoperative prediction of lymph node metastases in early TGCTs.

**Abstract:**

Accurate prediction of lymph node metastasis (LNM) in patients with testicular cancer is highly relevant for treatment decision-making and prognostic evaluation. Our study aimed to develop and validate clinical radiomics models for individual preoperative prediction of LNM in patients with testicular cancer. We enrolled 91 patients with clinicopathologically confirmed early-stage testicular cancer, with disease confined to the testes. We included five significant clinical risk factors (age, preoperative serum tumour markers AFP and B-HCG, histotype and BMI) to build the clinical model. After segmenting 273 retroperitoneal lymph nodes, we then combined the clinical risk factors and lymph node radiomics features to establish combined predictive models using Random Forest (RF), Light Gradient Boosting Machine (LGBM), Support Vector Machine Classifier (SVC), and K-Nearest Neighbours (KNN). Model performance was assessed by the area under the receiver operating characteristic (ROC) curve (AUC). Finally, the decision curve analysis (DCA) was used to evaluate the clinical usefulness. The Random Forest combined clinical lymph node radiomics model with the highest AUC of 0.95 (±0.03 SD; 95% CI) was considered the candidate model with decision curve analysis, demonstrating its usefulness for preoperative prediction in the clinical setting. Our study has identified reliable and predictive machine learning techniques for predicting lymph node metastasis in early-stage testicular cancer. Identifying the most effective machine learning approaches for predictive analysis based on radiomics integrating clinical risk factors can expand the applicability of radiomics in precision oncology and cancer treatment.

## 1. Introduction

Testicular germ cell tumours (TGCTs) are the most common form of cancer in males between 15 and 40 years, and its incidence is increasing all over the world [1,2,3,4,5]. With a cure rate of >95% in all patients and approximately 90% in patients with metastatic disease, TGCTs are now considered a curable cancer [6,7].

In the past, the majoritiy of TGCT patients with metastatic disease had a poor prognosis. However, the introduction of cisplatin-based chemotherapy regimens and more refined surgical techniques has led to a radical change in the prognosis of TGCTs [8]. Nowadays, only about 5% of patients die of metastatic disease due to cisplatin resistance [6]. 

There has been a steady increase in the incidence of TGCTs in some regions over the past 30 years, with the highest rates being in the Scandinavian countries, western and central Europe, the USA, Canada, Australia and Japan [9,10]. European countries will experience an increasing incidence burden from 2010 to 2035, with Baltic and Eastern European countries expected to see the largest increase [11]. The reasons are currently unknown. A study in Denmark has reported that the first generation of immigrants has a lower incidence of testicular cancer than the second generation, which may indicate the influence of environmental factors [12].

The prognostication of TGCT, of which the main histological types are seminoma and nonseminoma, the latter including pure nonseminoma and mixed germ cell tumours, has progressed considerably over the previous thirty years. Since the introduction of the International Germ-Cell Cancer Collaborative Group (IGCCCG) in 1997, there has been a widely accepted risk stratification model for metastatic disease. The determinants of poor prognosis were identified as non-pulmonary visceral metastases, amplification of the serum tumour markers HCG and AFP, and primary mediastinal nonseminoma and patients have been stratified into good, intermediate and poor risk categories [13]. After effective international collaboration, the IGCCCG guidelines were updated in 2021, including merged databases with data on 12,179 patients with metastatic germ cell tumours [6,7]. This has led to improved outcomes in TGCTs compared to the original IGCCCG classifications, with an increase in the five-year overall survival for nonseminoma from 92% to 96% in the good risk group, from 80% to 89% in the intermediate risk group, and from 48% to 67% in the poor risk group. For seminoma, the improvement in the five-year overall survival was from 86% to 95% in the good risk group and from 72% to 88% in the intermediate risk group [6,7].

If the physical examination and/or scrotal ultrasonography reveals a testicular tumour, the initial treatment is inguinal orchiectomy. Furthermore, clinical practice guidelines recommend the measurement of human chorionic gonadotropin (hCG and/or hCG-beta) for the management of TGCT, with differences depending on the histological type. Measurement of the serum concentrations of hCG and lactate dehydrogenase before and after orchiectomy is recommended for patients with pure seminoma and non-seminomatous TGCTs, with the addition of alpha-fetoprotein (AFP) in the latter case. In addition, in non-seminomatous TGCTs, hCG testing should be requested for staging and prognostic purposes prior to chemotherapy and/or additional surgery and to monitor response and early recurrence after therapy [14,15,16,17,18]. 

Due to the predictable pattern of the spread of TGCTs, metastatic lesions can be reliably identified in the diagnostic workup, with conventional computed tomography remaining the standard imaging modality despite numerous attempts to improve imaging with innovative techniques [14,16,19]. As we are focusing on early-stage TGCT patients (stage I) in our study, it is important to note that FDG PET/CT has similar sensitivity and specificity to conventional computed tomography in the identification of stage I non-seminomas; however, the sensitivity is insufficient to identify patients at a high risk of recurrence [20,21].

Although the serum tumour markers total beta-HCG and AFP are widely used, they lack specificity, and their diagnostic ability may depend on several factors, including the type of assay used and the upper reference level of the determination [22,23,24]. 

In addition, conditions such as liver disease or genetic factors can cause these markers to be falsely elevated [25]. 

To date, imaging, tumour markers and clinical nomograms are unreliable in predicting lymph node metastasis in TGCTs [26,27]. A recent study showed the high potential of images and radiomics in assessing the metastasis level [28]. However, focusing on feasibility, powerful modern classification algorithms were not investigated. 

Significant advances have been made in medical imaging by integrating high-resolution imaging, advanced computing technologies and artificial intelligence (AI). This integration has paved the way for the emerging field of radiomics [29,30]. 

Radiomics enables the identification and extraction of specific diagnostic image patterns, which are then transformed into quantifiable and analysable ‘big data’ through data characterisation algorithms [31,32]. In precision medicine, AI image analysis can help identify important image details that human radiologists may miss, offering repeatable and accessible ways to assess challenging lesions in the body to improve the detection, classification and monitoring of both the primary tumour and its associated metastases in various cancers, such as gastric, rectal and bladder cancers [33,34,35,36]. 

Our presented study combines machine learning-based radiomics with clinical predictors to improve the accuracy of predicting lymph node metastasis in early-stage TGCT patients. This advancement has the potential to significantly improve the accuracy of imaging in the clinical oncology setting, outrunning previous results.

## 2. Materials and Methods

### 2.1. Patients 

This retrospective study includes retroperitoneal lymph nodes of early-stage testicular cancer patients from January 2006 to December 2016. 

A comprehensive review of electronic medical records and the radiology information system was used to collect patient demographic, laboratory and clinical data. Incomplete clinical or imaging records, as well as missing histological confirmation following surgery were used as exclusion criteria. 

The study’s primary objective was to investigate retroperitoneal LN metastases from TGCT using clinical and imaging studies retrieved from electronic medical records. From an initial screening of 167 patients, only 91 patients met the selection criteria and were part of the final cohort. The recruitment pathway is shown in Figure 1.

### 2.2. Image Acquisition and Segmentation

All patients underwent contrast-enhanced CT according to standard clinical scanning protocols (detailed scanning parameters are provided in Appendix A). The images were acquired as part of the routine staging procedure prior to orchiectomy to determine the status of the disease (±2 weeks, mean time 3 ± 11 days, range 2 to 24 days). 

For image segmentation and analysis, all reconstructed images were retrieved from the hospital’s picture archiving and communication system (PACS).

### 2.3. Segmentation and Radiomics Feature Extraction

The evaluation of image features, such as histogram features or features from the co-occurrence matrix, was first introduced by Haralick et al. in 1973 [37], and has shown considerable potential in different types of cancer and for different types of questions [30,38]. 

Two experienced radiologists (>10 years’ experience in interpreting CT scans and with a strong background in texture analysis) contoured three retroperitoneal infrarenal lymph nodes per patient using Mint Lesion software (v3.8.4). The regions of interest (ROIs) were drawn on the CT images along the lesion contour on each successive slice within the boundaries of the retroperitoneal lymph node, excluding adjacent vessels, fat and normal tissue. The flowchart illustrating the ROI segmentation and feature extraction used to develop the model is shown in Figure 2.

Texture feature descriptors were used according to the guidelines of the Image Biomarker Standardisation Initiative (IBSI) [32]. A total of 85 image features were extracted from each of the 273 ROIs, covering the size and shape of the lymph node in three dimensions. In addition, the distribution of voxel intensities within the ROI was described using first-order statistics. Texture-based features were calculated from the grey level co-occurrence matrix (GLCM) to capture voxel intensity patterns. A list of all features and parameters is provided in Appendix A. 

### 2.4. Development of the Predictive Machine Learning Models

Four classic machine learning algorithms were evaluated to identify the best radiomics model for predicting lymph node metastases in testicular cancer: Random Forest (RF), Light Gradient Boosting Machine (LGBM), Support Vector Machine Classifier (SVC) and K-Nearest Neighbours (KNN) classifiers. For each classifier, three models were constructed: radiomics-only, clinical-only and combined radiomics–clinical prediction model.

For radiomics-only, 85 image-derived features were used based on the good performance of the features in previous experiments (See Appendix A). For clinical-only, clinical factors known for predictive value in TGCT [39,40,41,42] were included, such as age, pre-orchiectomy serum tumour markers AFP and B-HCG, histotype (seminoma vs. nonseminoma) and body mass index (BMI). For combined models, all features from radiomics-only and clinical-only were used. 

All models were trained on the same training and test splits, with the training data being upsampled using SMOTE to account for class imbalance. All hyperparameters were optimized to maximize the area under the receiver operating characteristic curve (AUC–ROC) using a grid search in nested cross-validation [43] (details in Appendix A).

The predictive performance of each classifier for LN metastases was evaluated using receiver-operating characteristic (ROC) curve analysis. The clinical utility of the predictive models was assessed using decision curve analysis (DCA), which evaluates the net benefit of the predictive models at different cut-offs in the training population and compares the performance of the models.

The models were constructed using custom-developed software implemented using Python 3.8.5 and the Scikit-learn 0.23.3 package [44,45] (details in Appendix A).

## 3. Results

### 3.1. Clinicopathological Characteristics 

Age, AFP levels, HCG levels, histotype and body mass index (BMI) were not statistically significantly different between LNM-positive and LNM-negative patients. 

The baseline clinicopathological patients’ characteristics are listed in Appendix A.

### 3.2. Dataset Characteristics and Preprocessing 

Three infrarenal retroperitoneal lymph nodes were segmented per patient, yielding 273 sample instances. We used a group shuffle split to divide the data into training (70%) and test (30%) groups on a patient-by-patient basis. This patient-by-patient split ensured that the lymph nodes of a given patient would remain together in either the training set or the test set. A tenfold cross-validation checked the robustness of the procedure. 

There were 33 instances in the category “relapse of disease in terms of lymph node metastases” (minority class) and 240 instances in the category “without relapse of disease” (majority class). 

Due to class imbalance in the dataset, we used the SMOTE oversampling technique to balance the data [46]. The balanced data were used as input variables for the machine learning modelling. 

### 3.3. Performance Evaluation of the Prediction Models

Eighty-five CT-derived radiomic features were fed into the machine learning models using RF, LGBM, SVC and KNN. A list of all features and parameters is provided in Appendix A. 

Based on the Random Forest algorithm, the combined clinical–radiomics model showed the best predictive performance with an AUC of 0.95 (±0.03 SD; 95% CI), accuracy of 87%, precision of 89%, recall of 86% and F1 score of 87%. 

The second-best performer was the model based on the Light Gradient Boosting Machine algorithm with an AUC of 0.93 (±0.05 SD; 95% CI), accuracy of 83%, precision of 87%, recall 80% and F1 score of 82%. Details of the performance of the radiomics-only, clinical-only and combined clinical–radiomics models of all classifiers are shown in Appendix A. To see how close the predictions of the two approaches are, see Figure 3 with the merged confusion matrix of the Random Forest-based models using radiomics-only and clinical-only values for classification.

Figure 4 shows the receiver operating characteristic (ROC) curves for the clinical, the radiomics and the combined clinical–radiomics models based on Random Forest algorithms.

Figure 5 shows the receiver operating characteristic (ROC) curves for the clinical, the radiomics and the combined clinical–radiomics models for the Random Forest (RF), Light Gradient Boosting Machine (LGBM), Support Vector Machine Classifier (SVC) and K-Nearest Neighbours (KNN). 

A decision curve analysis was performed to assess the clinical utility of the clinical–radiomics model combination. Figure 6 shows the net utility versus threshold probability trade-offs between true positives and false positives.

## 4. Discussion

Testicular germ cell tumours (TGCTs) are distinguished from other types of cancer by their unique patient population and high treatment success rates, representing an outstanding achievement in cancer treatment [8,47].

In addition to cure, minimising the immediate and long-term side effects of treatment is the main goal. This is particularly important given the young age of patients and their longevity after cure [48,49,50,51,52,53].

Medical imaging has greatly advanced cancer diagnosis and treatment planning with the emergence of ‘radiomics’, a field that involves high-throughput data mining of medical images. Radiomics has significant potential to improve clinical decision support in cancer care by providing a non-invasive and cost-effective approach [30,31]. As radiomics deals with large amounts of medical image data (“big data”), efficient methods are needed to extract relevant information from these large radiomic datasets [29]. 

In this study, we analysed the performance of different machine learning methods, namely Random Forest (RF), Light Gradient Boosting Machine (LGBM), Support Vector Machine Classifier (SVC) and K-Nearest-Neighbours (KNN), in predicting lymph node metastases in patients with early-stage testicular germ cell tumours (TGCTs). 

We constructed radiomics-only, clinical-only and combined predictive models for each classifier, integrating clinical and radiomic features to identify patients who require adjuvant therapy and those who do not. 

Our main findings can be summarised by the following: 

The combined radiomics–clinical model based on the Random Forest algorithm showed the best predictive performance with an AUC of 0.95 (±0.03 SD; 95% CI) and an accuracy of 87%, indicating that the addition of clinical features improved the predictive performance (accuracy of the radiomics-only model 85% vs. the clinical only model 79%). 

The Light Gradient Boosting Machine classifier performed second best with an AUC of 0.93 (±0.05 SD; 95% CI) and an accuracy of 83%. In contrast to the Random Forest, adding clinical features to the radiomics prediction model worsened the predictive performance (accuracy of the radiomics-only model 85% vs. the clinical-only model 73%).

In our analysis, models based on the Support Vector Machine Classifier and K-Nearest Neighbours performed significantly worse than Random Forest and Light Gradient Boosting Machine. However, it is worth noting that the combined radiomics–clinical model outperformed the radiomics-only prediction models in both cases.

This is in line with the results of our previous study [28], in which we demonstrated that logistic regression analysis is useful for the prediction of lymph node metastasis in TGCT patients, with the best predictive performance being the combined clinical–radiomics model with an AUC of 0.95 (±0.03 SD; 95% CI).

The serum biomarkers AFP, β-HCG and LDH play an important role in the diagnosis and prognosis of TGCTs, and they are included in the prognostic index of the International Germ Cell Cancer Consensus Group [54]. However, their sensitivity is limited, as around 40% of men have ‘normal’ levels at recurrence [55]. Studies have suggested additional prognostic factors such as age and BMI, but their role is unclear and continues to be debated [39,40,41,42].

So far, there is no evidence that imaging, preoperative serum tumour markers or clinical nomograms can reliably predict nodal involvement [26,27]. Inadequate management of TGCT patients places them at risk of adverse outcomes, as both overtreatment and undertreatment carry equal risks.

There are few studies on discriminating between benign and malignant LNs in testicular cancer. Baessler et al. [56] found that a CT radiomics-based machine learning classifier could predict lymph node histopathology after dissection following chemotherapy in metastatic non-seminomatous TGCT patients. They used a Support Vector Machine learning classifier in their single-centre retrospective study of 80 patients and 204 lesions. The model distinguished between benign and malignant histopathologies with an accuracy of 81%. 

Nevertheless, in contrast to our study, they did not include clinical factors in their radiomics approach to improve diagnostic accuracy. They also split their moderate-sized dataset into three subgroups: 63 patients were assigned to train and only 19 to test. The splitting of data for validation purposes is common practice. However, doing so reduces statistical power because the sample sizes in both groups are smaller than in the initial sample. 

To overcome this problem, we used a cross-validation technique that uses multiple data splits to avoid overfitting while still providing accurate estimates of the model coefficients [57].

We are confident that our combined prediction model will generalise better to novel data due to our tenfold cross-validation approach, the a priori inhomogeneity of our dataset and the integration of clinical risk factors. Therefore, future prospective studies should be conducted to validate our trained model further.

Several clinical models have been developed to predict the dignity of LN metastases. Nevertheless, these models have shown inconsistent results and have not yet been adopted for clinical decision-making [26,58,59].

In summary, the identification and implementation of novel biomarkers may be helpful for early diagnosis and disease recurrence monitoring. 

The present study, however, has some limitations that we acknowledge. First, the study’s retrospective nature and the small cohort sizes might have led to unavoidable selection bias. 

Secondly, two different scanners were used to acquire the CT images. Thirdly, the results of this study were obtained from a single centre. Due to the high cure rate of stage I TGCTs, it is challenging to power studies in a way that allows for prognostic and predictive factors to be adequately investigated. Therefore, prospective and multicentre validation is warranted to provide higher-level evidence in the following studies. 

Fourth, due to the small sample size and relapse events, we could not include the classical prognostic pathohistological factors (primary tumour size and rete testis invasion for seminoma and lymphovascular invasion and presence of embryonal carcinoma for nonseminoma) in our analysis. Their inclusion in the combined radiomics–clinical model may have further improved accuracy and is a promising addition for future validation studies.

Finally, in addition to protein-based tumour markers, non-coding RNAs, especially stem cell-associated microRNAs such as miR-371a-3p and miR-302/367 clusters, show superior sensitivity compared to traditional markers in the detection of newly diagnosed TGCT patients, demonstrating their potential as biomarkers [60,61]. 

## 5. Conclusions

In summary, our combined Random Forest-based radiomics–clinical model represents an exciting tool for better prediction of lymph node involvement in early-stage TGCTs, with the potential to reduce over- and undertreatment in this young patient population. Further validation in larger prospective clinical trials should combine this approach with novel clinical biomarkers.

## Figures and Tables

**Figure 1 cancers-15-05630-f001:**
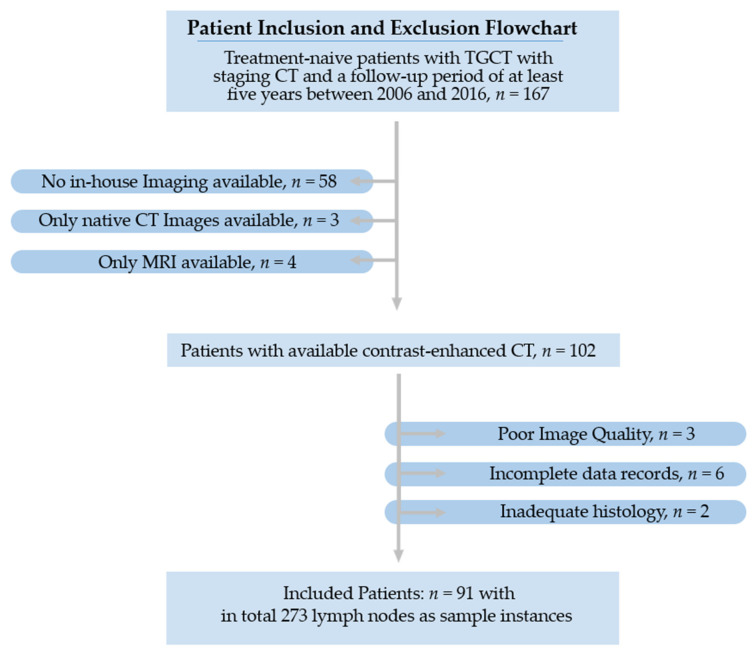
Recruitment pathway.

**Figure 2 cancers-15-05630-f002:**
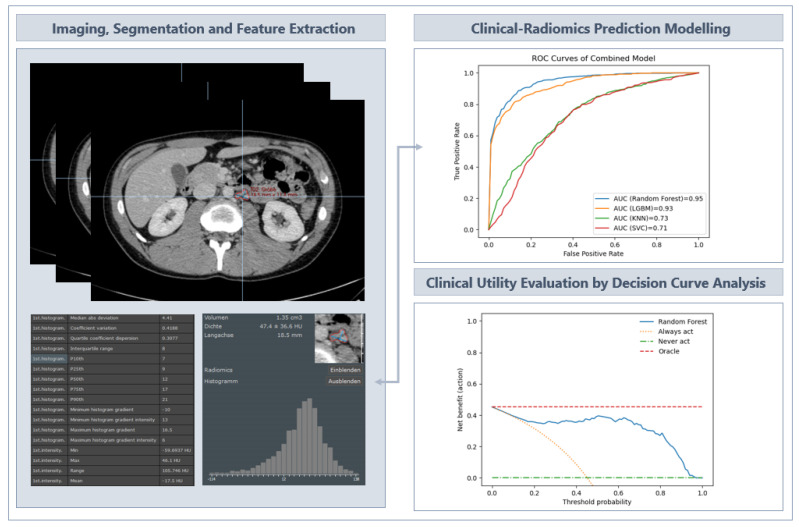
Schematic of the ROI segmentation and feature extraction process that provided the input for the development of the models, with decision analysis as the clinical utility evaluation tool. LGBM = Light Gradient Boosting Machine; KNN = K-Nearest Neighbours; SVC = Support Vector Machine Classifier.

**Figure 3 cancers-15-05630-f003:**
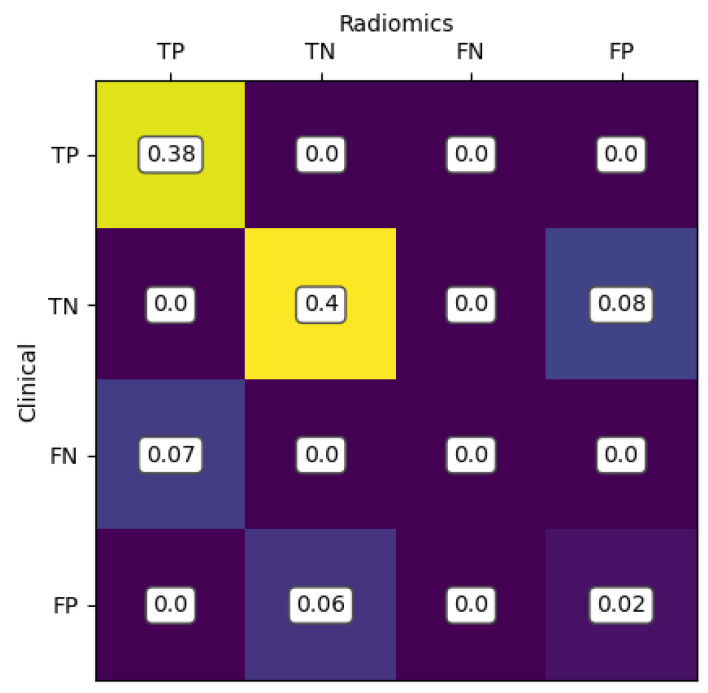
The most values are aligned along the main diagonal, indicating a high level of agreement between clinical and radiomic results. In particular, 38% are correctly classified as true positives (TP) and 40% as true negatives (TN) by both models. Both models misclassify 2% as false positive (FP). In addition, the radiomics model misclassifies 8% of cases as FP, but the clinical model correctly identifies these. In contrast, the clinical model misclassified 7% FN and 6% FP.

**Figure 4 cancers-15-05630-f004:**
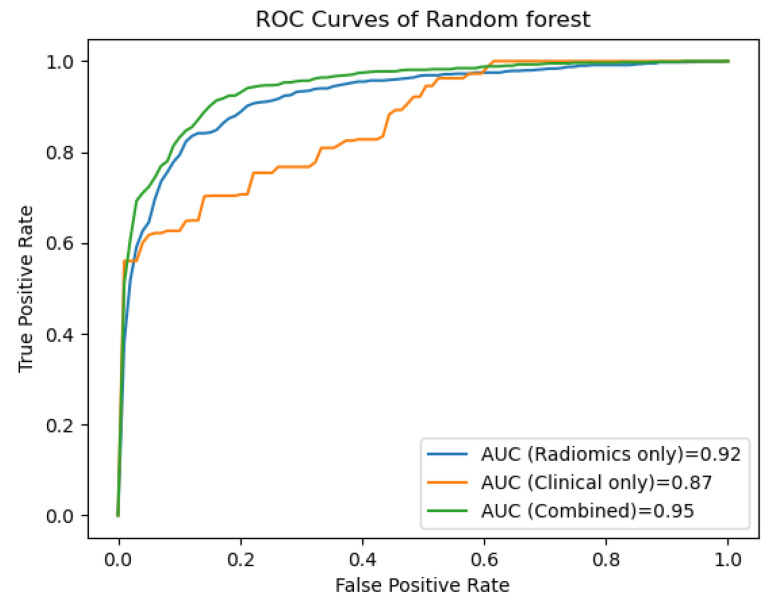
The ROC curves of the Random Forest-based prediction models show that the combined model outperforms the radiomics-only and the clinical-only model in predicting lymph node metastasis (95% vs. 92% and 88%, respectively).

**Figure 5 cancers-15-05630-f005:**
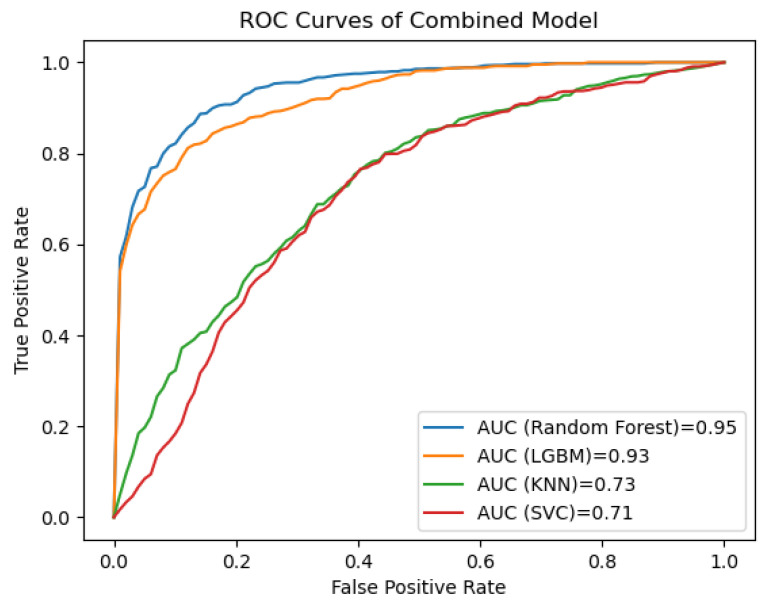
The ROC curves of the prediction models show that the combined RF-based model outperforms the LGBM-based model. In our analysis, SVC- and KNN-based prediction models performed considerably worse. LGBM = Light Gradient Boosting Machine; KNN = K-Nearest Neighbours; SVC = Support Vector Machine Classifier.

**Figure 6 cancers-15-05630-f006:**
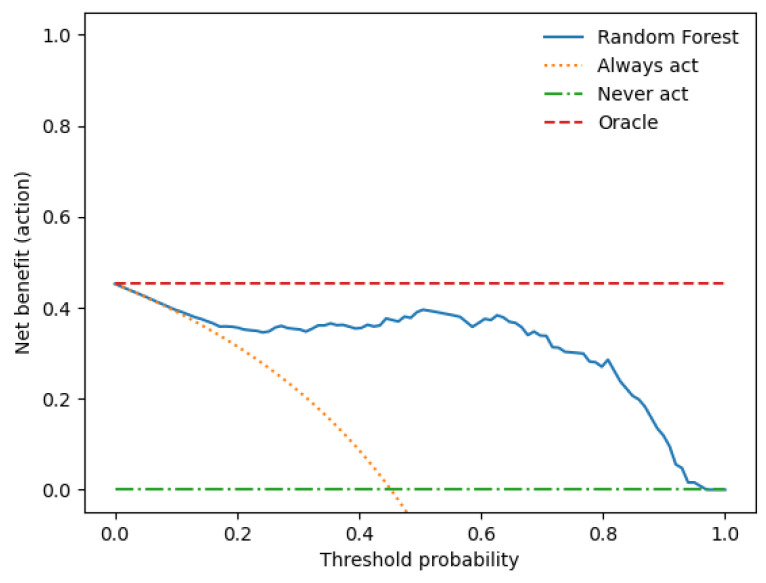
The decision curve shows that when the threshold probability is between 0 and 0.89, the use of the Random Forest-based combined prediction model provides an increased benefit over treating all or none of the patients.

## Data Availability

Data are contained within the article and Appendix A.

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
