# Peer review of "Radiomics and Clinicopathological Characteristics for Predicting Lymph Node Metastasis in Testicular Cancer"

_cancers, 2023, doi:10.3390/cancers15235630_

Round 1

Reviewer 1 Report

Comments and Suggestions for Authors

1)"There are two main histological types: seminoma and non-seminoma,with the latter including both pure non-seminoma and mixed germ cell tumours." A reference is lacking I recommend to quote the following paper reporting the relation between HCG total and/or freeBHCG and the hystologic type. 

Ferraro S, et al. Human Chorionic Gonadotropin Assays for Testicular Tumors: Closing the Gap between Clinical and Laboratory Practice. Clin Chem 2018;64:270-278

2) It has to be reported data about worldwide mortality/ morbility rates, trends and incidence of advanced stages at diagnosis.Epidemiological data in the first lines of the introduction are not enough.

3)German guidelines recommend ....Report about the recommendation of international guidelines.

4) "Although the common tumour markers AFP, beta-HCG and LDH are widely used, they lack specificity". Total HCG is the main relevant biomarker together with AFP, and the diagnostic capability may depend on several factors including, the type of assay and the determination upper reference level 

To improve this concept read and quote this paper: "A step forward in identifying the right human chorionic gonadotropin assay for testicular cancer. Clin Chem Lab Med"

5) Report the acronyms under the figure 2 or the AUCs are difficult to understand.

6)"The baseline clinicopathological characteristics of the patients are listed in supplemental table S3)".The baseline clinicopathological characteristics is the first and basic results which should be reported in Table1 and not in the supplementary material.

7)The variables entering in the model how have been selected?Univariate analysis? these are reported in the Supplementary, I strongly suggest to report here the variables in the results. 

8)87% (AUC 0.95±0.03; 95% CI). 0.03 is the SD, report the 95%CI

9)Acronyms have to be spelled out the first time are used, and not at the end in the discussion." forest (RF), light gradient boosting machine (LGBM), support vector

machine classifier (SVC) and k-nearest neighbour classifier (KNN)"

10) In the discussion declare the study limitations. the sample size might be a limitation, so that providing CI may be useful to consider the precision of the estimates.

11) I recommend to highlight in the discussion that this is only a preliminary evaluationdue to the low sample size.

Comments on the Quality of English Language

Required extensive editing

Author Response

Dear Reviewer,

On behalf of my co-authors, I would like to thank you for taking the time to review and comment upon our manuscript entitled "Radiomics and Clinicopathological Characteristics for Predicting Lymph Node Metastasis in Testicular Cancer" (ID: cancers-2677934).

Below, I provide the point-by-point responses.

Thank you again for your thoughtful comments.

On behalf of all the co-authors

Sincerely,

Catharina Lisson

Senior physician and clinician scientist, Department of Diagnostic and Interventional Radiology,

University Hospital of Ulm, Albert-Einstein-Allee 23, 89081 Ulm, Germany

(+) 49 (0) 731 500 61171

[email protected]

Reviewer 2 Report

Comments and Suggestions for Authors

Thank you for giving me the opportunity to review this article.

The authors investigated the effectiveness of CT radiomics and whether these methods improved the identification quality of suspected lymph nodes in patients with early-stage testicular cancer. AI-based reading is a very interesting field because it has the potential to reduce human error and increase the detection rate of suspicious lesions.

The focus of this study is impressive, but I think some additional explanations are necessary.

-The authors have submitted the following paper to "Onco," which has already been published. The study participants in that research are the same as those in this study. The methods differ slightly, but the authors' focus seems almost identical. I need clarification on why these two studies are required to be submitted separately. Please describe the originality of this study compared to the other one.

CT Radiomics and Clinical Feature Model to Predict Lymph Node Metastases in Early-Stage Testicular Cancer

Onco 2023, 3(2), 65-80; https://doi.org/10.3390/onco3020006

-Please describe the procedure for statistical analysis in the methods.

Author Response

(The authors gave the same response as above.)

Round 2

Reviewer 1 Report

Comments and Suggestions for Authors

No further comments.